# Targeting circDGKD Intercepts TKI’s Effects on Up-Regulation of Estrogen Receptor β and Vasculogenic Mimicry in Renal Cell Carcinoma

**DOI:** 10.3390/cancers14071639

**Published:** 2022-03-23

**Authors:** Jie Ding, Xin-Gang Cui, Hao-Jie Chen, Yin Sun, Wei-Wei Yu, Jie Luo, Guang-Qian Xiao, Chawnshang Chang, Jun Qi, Shuyuan Yeh

**Affiliations:** 1Urology Department, Xin Hua Hospital Affiliated to Shanghai Jiao Tong University School of Medicine, Shanghai 200092, China; dingjie@xinhuamed.com.cn (J.D.); cuixingang@xinhuamed.com.cn (X.-G.C.); xiaokeai945@sjtu.edu.cn (H.-J.C.); 2George H. Whipple Lab for Cancer Research, Departments of Urology, Pathology and the Wilmot Cancer Institute, University of Rochester Medical Center, Rochester, NY 14642, USA; yin_sun@urmc.rochester.edu (Y.S.); weiwei_yu@urmc.rochester.edu (W.-W.Y.); luojie@med.umich.edu (J.L.); chang@urmc.rochester.edu (C.C.); 3General Department of Tongji Hospital, Tongji Medical College of Huazhong University of Science and Technology, Wuhan 430030, China; 4Department of Pathology and the Wilmot Cancer Center, University of Rochester Medical Center, Rochester, NY 14642, USA; guangqian_xiao@urmc.rochester.edu

**Keywords:** estrogen receptor β, vasculogenic mimicry, renal cell carcinoma, circular RNA, VE-cadherin

## Abstract

**Simple Summary:**

Our aim was to elucidate the molecular mechanisms of how tyrosine kinase inhibitors, including sunitinib, contribute to vasculogenic mimicry (VM) formation and progression in renal cell carcinoma. We demonstrated that sunitinib and axitinib could induce ERβ expression in RCC cell lines and ERβ transcriptionally up-regulated the circular RNA of DGKD (circDGKD, hsa_circ_0058763) expression. The circDGKD could sponge tumor suppressor miR-125-5p family members and consequently led to increased VE-cadherin, the key adhesion molecule in the VM formation process, which is targeted by miR-125-5p at the 3′ UTR, providing novel targets for combination therapy in clinical metastatic RCC patients.

**Abstract:**

Vasculogenic mimicry (VM) has been reported as an alternative channel to increase tumor nutrient supplies and accelerate tumor progression, and is associated with poor survival prognosis in multiple cancers, including renal cell carcinoma (RCC). The currently used anti-angiogenic treatment for metastatic RCC, sunitinib, a tyrosine kinase inhibitor (TKI), has been reported to induce VM formation. Previously we identified that the estrogen receptor β (ERβ) functions as an oncogenic factor to promote RCC progression, supported by the analytic results from The Cancer Genome Atlas (TCGA) database. We have also found evidence that sunitinib induces RCC VM formation by up-regulating ERβ expression. In this study, we further demonstrated that treatment with sunitinib, as well as axitinib, another TKI, could induce ERβ expression in RCC cell lines. Clinical clear cell RCC (ccRCC) patients with higher ERβ expression are more likely to be found VE-cadherin positive and VM positive. Mechanism dissection showed that TKI- induced ERβ transcriptionally up-regulates the circular RNA of DGKD (circDGKD, hsa_circ_0058763), which enhances VE-cadherin expression by sponging the microRNA miR-125-5p family. Targeting circDGKD intercepts sunitinib-pretreatment-induced RCC VM formation, reduces metastases and improves survival in an experimental orthotopic animal model. Targeting ERβ/circDGKD signals may improve the TKI efficacy and provide novel combination therapies for metastatic RCC.

## 1. Introduction

Cancer patient deaths primarily result from tumor recurrence and metastases that are resistant to conventional therapies. The accepted tenet underlying tumor survival has been that blood and nutrient supplies are required to sustain tumor growth and metastases [1]. This important premise ignited the field of neoplastic angiogenesis research, which focused on targeting endothelial cells forming the neovasculature of growing tumors and served as the major organizing principle for drug discovery and clinical trials. 

Kidney cancer accounts for at least 3% of malignant diseases [2]. The incidence and mortality of renal cell carcinoma (RCC), the major kidney cancer, appears to be rising. Approximately 70,680 new cases were estimated in 2021 in the United States, resulting in more than 13,780 deaths, mainly due to metastatic disease [3]. RCC is divided into three major histopathologic groups—clear cell (ccRCC), papillary (pRCC), and chromophobe RCC (chRCC), as well as some rare histopathologic entities. The histopathology is of crucial relevance for determining treatment strategies, including drug sequencing in RCC patients, especially in a metastasized situation [4]. RCC is a type of highly vascularized solid tumor [5], and anti-angiogenic tyrosine kinase inhibitors (TKIs) such as sunitinib [6] and pazopanib [7] had been the standard therapy for ccRCC, which has constituted around 80% of all RCC for many years. Axitinib is a new TKI also prescribed as a first-line treatment for metastatic ccRCC patients [8]. However, the clinical outcome after receiving those anti-angiogenic therapies, although improved significantly compared to placebo, failed to live up to expectations. This led us to wonder whether there might be an alternative source of blood and nutrient supply. Because even with the theories of epithelial–mesenchymal transition (EMT) or cancer stem cell (CSC) development [9], the demand for blood and nutrient supplies are critical to supporting the astonishing growth rate and ultimate huge size of RCCs. 

Angiogenesis, the sprouting of new blood vessels from an existing vasculature, is a main driving force for the malignancy and progression of many types of tumors, and, therefore, is considered an essential pathologic feature of cancer as well as playing a key role in tumor dissemination and metastasis [10]. One of the alternative perfusion paradigms is “vasculogenic mimicry”, also referred to as “vascular mimicry” (VM). The initial report and molecular characterization of VM was made in melanoma [11]. Later, VM was also found in breast cancer [12], hepatic carcinoma [13], colorectal cancer [14], gastric cancer [15,16], ovarian cancer [17], astrocytoma [18], mesothelial sarcoma [19], etc. Interestingly, these findings agree with very early reports by others suggesting the perfusion of tumors via nonendothelial-lined channels. Furthermore, there is increasing evidence that tyrosine kinase inhibitors (TKIs), such as sunitinib, could promote VM formation, implicating the strong stimulus of tumors striving for nutrient supplies in the face of angiogenesis inhibition [20,21,22]. 

Vascular endothelial (VE)-cadherin, also known as CD144, encoded by the human gene CDH5, is a classical member of the cadherin superfamily. The encoded protein is a calcium-dependent cell–cell adhesion glycoprotein composed of five extracellular cadherin repeats, a transmembrane region, and a highly conserved cytoplasmic tail [23]. Functioning as a classic cadherin by imparting to cells the ability to adhere in a homophilic manner, the protein plays an important role in endothelial cell biology and morphogenesis through control of the cohesion and organization of the intercellular junctions [24]. In melanoma, VE-cadherin was exclusively expressed by highly aggressive tumor cells and was undetectable in the less aggressive tumor cells, suggesting the possibility of a vasculogenic switch [25]. The down-regulation of VE-cadherin expression in the aggressive melanoma cells abrogated their ability to form vasculogenic networks and directly tested the hypothesis that VE-cadherin is critical in melanoma VM formation. The critical role of VE-cadherin was not only reported in melanoma but also in other cancers [26,27,28].

In previous studies, we have identified that ERβ functions as an oncogene in RCC, promoting progression via regulating the HOTAIR-associated competitive endogenous RNA (ceRNA) network [29]. We have also found evidence that sunitinib treatment induces RCC VM formation by up-regulating ERβ expression [30]. In this study we further demonstrated that sunitinib and even axitinib could induce ERβ expression in RCC cell lines. Clinical ccRCC patients with higher ERβ expression are more likely to be found VE-cadherin positive and VM positive. Mechanism dissection showed that TKIs could induce ERβ transcriptionally to up-regulate the circular RNA of DGKD (circDGKD, hsa_circ_0058763), which could then sponge miR-125-5p family members and consequently lead to increased VE-cadherin, the key adhesion molecule in the VM formation process. Targeting circDGKD intercepts sunitinib-pretreatment-induced RCC VM formation, reduces metastases, and improves survival in an experimental orthotopic animal model, providing novel targets for combination therapy in clinical metastatic RCC patients.

## 2. Results

### 2.1. VM Formation in RCC Induced by Sunitinib Treatment via ERβ Up-Regulation Could Be Blocked by Silencing VE-Cadherin

First, we treated RCC cells with 1 μM sunitinib, which is a widely used treatment for metastatic ccRCC patients, for one week as we previously reported [30] and repeated the in vitro 3D culture system to conduct tube formation assays and evaluate the VM potential. The ability of A498 (Figure 1a) and SW839 (Figure 1b) cells to form polygonal structures, both by morphology recognition and by tubule number and area quantification, were significantly enhanced. We repeated the detection of ERβ levels by Western blot in 786-O and A498 cells, which we found to be responsible for the enhanced VM capability in our previous study. The result showed increased levels of ERβ in both 786-O and A498 cells after sunitinib treatment (Figure 1c). Meanwhile, we examined the effects of axitinib, which is also a first-line treatment for high-risk metastatic ccRCC patients. After 1 μM axitinib treatment, the ERβ expressions were also increased in A498 and 786-O cells (Figure 1c). We next analyzed the TCGA ccRCC cohort data and found ERβ (ESR2) negatively correlated with patients’ overall and disease-free survival (Figure 1d,e). The knockdown of ERβ (shERβ) in 786-O (Figure 1f) and Caki-1 (Figure 1g) cells severely debilitated their ability to form tubules. Based on RNA-Seq experiments we conducted previously [29], we analyzed the expression levels of 20 genes that had been reported to regulate VM formation and found SLPI, VE-cadherin (CDH5), and Vimentin (VIM) were among the factors that changed accordingly and relatively more significantly (Figure 1h). As shown by qPCR analysis, the overexpression of ERβ (oeERβ) increased SLPI, VE-cadherin, and Vimentin (Figure 1i), and the knockdown of ERβ decreased all three of them (Figure 1j). For the next step, we constructed lentiviral shRNA for SLPI (shSLPI), VE-cadherin (shCDH5), and Vimentin (shVIM) to compare their relative potency in the modulation of VM (Figure 1k). In the 786-O cell line, which has a high endogenous ERβ expression and strong endogenous tube formation capability, silencing VE-cadherin exhibited the most obvious effect in inhibiting tube formation. We then conducted IHC staining in ccRCC patients’ samples (Figure 1l) and found that the patients with higher ERβ staining (+/++) were more likely to be VE-cadherin (+) (Figure 1m, *p* < 0.01) and VM (+) (Figure 1n) than the patients with lower ERβ (−/±) staining. Supportively, knocking down VE-cadherin significantly interrupted the sunitinib-treatment-enhanced tube formation in A498 cells (Figure 1o). 

### 2.2. The miRNA-125-5p Is Involved in Mediating ERβ’s Modulation of VE-Cadherin

To determine how ERβ regulates VE-cadherin mRNA, 786-O cells were starved of estrogen contact by incubating them in 5% charcoal-dextran-stripped (CDS) FBS DMEM media for 4 days, then treated with 10 nM estradiol (E2) for 6, 12, 48, or 72 h. The qPCR results showed that mRNA of VE-cadherin did not increase in 6, 12, or even 24 h, whereas it significantly increased at 48 and 72 h (Figure 2a). We speculate that the regulation might be indirect post-transcriptional regulation, and ERβ may function via non-coding RNA regulatory mechanisms. We hypothesize that miRNAs, which have been reported to down-regulate mRNA post-transcriptionally [31], may be involved. To test the hypothesis, we cloned the 3′ UTR of VE-cadherin (CDH5) into psiCHECK2 luciferase plasmid (Figure 2b). The relative luciferase activity decreased significantly after knocking down ERβ (Figure 2c). Since miRNAs are known to be prevalent and important post-transcriptional regulators, we tried to sort out the key miRNAs that can bind to and target the 3′ UTR of VE-cadherin, as well as be regulated by ERβ in RCC. Using Targetscan, three miRNAs, including miR-27-3p, miR-101-3p, and miR-125-5p, have 8mer seed region matching. Based on a GEO RCC dataset (GSE37989), miR-125-5p, including 125a-5p and 125b-5p, and miR-27b-3p but not miR-27a-3p, were decreased in RCC samples and further decreased in metastatic RCC samples (Figure 2d). Based on a postulation algorithm that VE-cadherin-mediated VM formation would promote RCC progression, the miRNAs that target VE-cadherin and decrease in RCC (potential tumor suppressor), rather the ones that increase, might be the key candidate players (Figure 2e). Clinical data from the Starbase further confirmed that both miR-125a-5p and miR-125b-5p were significantly lower in RCC than normal tissues (Appendix A) so we focused on them. Based on bioinformatics prediction, miR-125-5p has two binding sites on the 3′ UTR of VE-cadherin, one is 7mer binding (684–690) and the other is 8mer binding (1080–1087) (Figure 2f). Further analysis showed a negative correlation between miR-125b-5p and VE-cadherin (CDH5) expression in clinical ccRCC samples (Figure 2g). Since miR-125a-5p and miR-125b-5p share the same seed region sequence, we just focused on miR-125b-5p to conduct the subsequent functional experiments. The overexpression of miR-125b-5p interrupted oeERβ-increased VE-cadherin mRNA (Figure 2h) and protein (Figure 2i) in A498 cells. The inhibition of miR-125b-5p rescued shERβ-decreased VE-cadherin mRNA (Figure 2j) and protein (Figure 2k) in 786-O cells. The overexpression of miR-125b-5p interrupted the oeERβ-enhanced tube formation in A498 cells (Figure 2l), whereas treatment with the miR-125b-5p inhibitor rescued shERβ-debilitated tube formation in 786-O cells (Figure 2m). The above results demonstrated that ERβ might regulate VE-cadherin via antagonizing miR-125-5p, which targets the 3′ UTR of VE-cadherin. 

### 2.3. ERβ Increases circDGKD Which Sponges miRNA-125-5p in RCC

A previous report suggested the regulation of the miRs may not always function through transcriptional regulation [32]. In recent years, circular RNA (circRNA) has been reported to function as a miRNA sponge and antagonize miRNA function [33,34,35]. Using Starbase, we conducted a circRNA screen and chose the top 10 circRNA candidates that might sponge miR-125-5p (Appendix A, the original 10th MLL3 was not selected due to database information mismatch between Starbase and Circbase, so the 11^th^ ELF2 was chosen to be among the top 10 candidates). Since one gene’s mRNA and circRNA are transcribed from the same genomic location, the change of mRNA might lead to the same changing trend of circRNAs. Analyzing the RNA-Seq data we previously conducted [29], we found that the mRNA of DGKD increased after E2 treatment for 24/72 h and decreased after knocking down ERβ (Figure 3a). The detailed information about circDGKIhsa_circ_0058763) was retrieved from CircBase (Figure 3b). The binding site information for miR-125-5p was retrieved from Starbase (Figure 3c). We designed primers specific for detection of circDGKD (Figure 3d) and performed the qPCR assay after RNase R digestion to confirm the authenticity of circDGKD (Figure 3e). Then, knockdown of ERβ in 786-O cells decreased circDGKD (Figure 3f), whereas overexpression of ERβ in A498 cells increased circDGKD significantly (Figure 3g). 

An RNA pull-down assay using a biotinylated probe specific for the junction region showed that both miR-125a-5p and miR-125b-5p were highly enriched in the pull-down product, indicating the specific interaction of miR-125-5p with circDGKD rather than the circPCMTD1 that served as a negative control (Figure 3h). A fluorescence in situ hybridization (FISH) assay using probe targeting circDGKD tagged with Cy3 and exogenous miR-125b-5p tagged with FAM also showed the co-localization of circDGKD and miR-125b-5p in 786-O and ACHN cells (Figure 3i), further confirming circDGKD sponges miR-125-5p.

### 2.4. Knockdown of circDGKD Reverses ERβ’s Modulation of VE-Cadherin and VM Formation

We constructed the shRNA plasmid of circDGKD targeting the unique junction region of circDGKD as well as wild-type (Wt)/MRE-125 mutant (Mut) circDGKD overexpression plasmid by adding the *Alu* element on the flank (Figure 4a). We tested the effect of these two plasmids using qPCR and the results showed circDGKD and VE-cadherin mRNA levels changed in 786-O and A498 cells (Figure 4b,c, respectively). Luciferase assays suggested that both shcircDGKD and miR-125b-5p decreased the luciferase activity of psiCHECK2-CDH5, yet the silencing of circDGKD could not further decrease the luciferase activity in cells that were transfected with miR-125b-5p (Figure 4d), also suggesting that circDGKD functions via interaction with miR-125. Overexpression of the circDGKD with MRE-125 Wt, instead of the MRE-125 Mut, significantly increased the luciferase activity of psiCHECK2-CDH5 in the absence or presence of miR-125 (Figure 4e). 

We then further confirmed the function of circDGKD. The overexpression of circDGKD restored shERβ-decreased VE-cadherin in both mRNA (Figure 4f) and protein (Figure 4g), and rescued the shERβ-impaired VM formation in 786-O cells (Figure 4j). Sunitinib treatment significantly increased VE-cadherin mRNA (CDH5, Figure 4h) and protein level (Figure 4i) in A498 cells, and boosted the VM formation capacity (Figure 4k), whereas this effect was interrupted by shcircDGKD. Together, results from Figure 4 further confirmed that ERβ up-regulates circDGKD to sponge and antagonize miR-125-5p, and consequently increases the VE-cadherin and RCC VM formation capacity.

### 2.5. ERβ Can Bind to the Promoter of DGKD and Transcriptionally Regulates circDGKD Expression

To further delineate the mechanism of ERβ-regulated circDGKD, we applied the qPCR assay to examine the mRNA and circRNA of DGKD, and showed that they both decreased after knocking down ERβ in 786-O cells, and increased after overexpressing ERβ in A498 cells (Appendix A). As a member of the nuclear receptor superfamily, ERβ may function via binding to the estrogen response elements (EREs) located on its target gene’s promoter region to modulate their expression. First, to test this hypothesis, we analyzed the 2 kb region of the DGKD promoter using ALGGEN-PROMO and found five putative EREs (Figure 5a). The ChIP assay results showed ERE4 to be the potential binding site (Figure 5b). Then, we cloned this 2 kb of DGKD promoter into pGL3-basic luciferase report vector (pGL3-wild-type) (Figure 5c), and then mutated the ERE4 into the BamHI cutting site (Figure 5d). The luciferase assay results showed that shERβ significantly decreased the DGKD-Wt promoter-luciferase reporter activity in 786-O cells (Figure 5e), and overexpression of ERβ significantly increased the luciferase activity in A498 cells (Figure 5f), whereas after mutation of the ERE4 (position −1203 to −1211), neither shERβ nor oeERβ could change the luciferase activity, indicating that ERβ modulated the DGKD mRNA/circRNA expression via binding to this ERE4 on its promoter.

RNA-Seq data were retrieved from the TCGA database using the UCSC Cancer Genome Browser (https://genome-cancer.soe.ucsc.edu/, accessed on 8 October 2021) and analyzed as we reported before [29]. The results revealed a positive correlation between the expression level of ERβ and DGKD (Figure 5g), and between DGKD and VE-cadherin (Figure 5h). In addition, we retrieved the single largest GEO RCC dataset accessible (GSE19949 from the Mayo Clinic), and the analyzed results also showed a positive correlation between ERβ and DGKD (Figure 5i), and between DGKD and VE-cadherin (Figure 5j). The data of normal kidney tissue from the GTEx database also demonstrated a positive correlation between ERβ and DGKD (Figure 5k), and between DGKD and VE-cadherin (Figure 5l). Furthermore, TCGA ccRCC data analysis showed a negative correlation between the expression level of DKGD and miR-125-b (Figure 5m), and between miR-125-b and VE-cadherin (Figure 5n), further supporting our hypothesis. Together, the results from Figure 5 demonstrated that ERβ binds to the promoter of DGKD and transcriptionally regulates circDGKD expression, and their positive correlation was further confirmed by the analyzed results from clinical databases. 

### 2.6. Orthotopic Mouse Model Studies Confirmed That shcircDGKD Intercepts Sunitinib-Pretreatment-Induced RCC VM Formation, Reduces Metastases, and Improves Survival

We established an orthotopic RCC model by implanting 786-O-Luc-Scr/786-O-Luc-shcircDGKD cells (1 × 10^6^), pretreated with or without sunitinib (1 μM) for 1 week, into subrenal capsules of female nude mice (Figure 6a). After tumor development, the mice were given 20 mg/kg DMSO or sunitinib once a day intraperitoneally. IVIS detections were recorded every 2 weeks for up to 8 weeks (Figure 6b), and CD31/PAS double staining identified the VM structures in tumor xenografts (Figure 6c). Among the mice, four out of eight in the vehicle/Scr group, and six out of eight in the STN/Scr group were VM (+), whereas none in the vehicle/shcircDGKD group and only one in the STN/shcircDGKD group were VM (+) (Figure 6d). IVIS showed that five mice in the vehicle/Scr group and seven mice in the STN/Scr group had metastatic foci, whereas all mice in the vehicle/shcircDGKD and STN/shcircDGKD groups were metastasis-free (Figure 6e). The shcircDGKD treatment significantly reduced metastatic foci numbers in both the vehicle and sunitinib-treated groups (Figure 6f), and prolonged survival length of nude mice (Figure 6g). This data demonstrates the potential of targeting circDGKD in improving the therapeutic effect of TKIs in metastatic RCC patients. The ERβ/circDGKD/miR-125-5p/VE-cadherin pathway diagram is summarized and illustrated in Figure 7. Sunitinib induces VM formation by up-regulating the ERβ/circDGKD/miR-125-5p/VE-cadherin axis, whereas the sunitinib plus shcircDKGD combo treatment suppresses angiogenesis and VM simultaneously.

## 3. Discussion

Vasculogenic mimicry (VM) is the formation of fluid-conducting channels by highly aggressive and genetically dysregulated tumor cells, which acts as a complemental source of blood and nutrient supplies. It is reasonable to assume that during anti-angiogenic regimen treatment, when neo-angiogenesis is under suppression, the tumor growth would be more dependent on the supplies from VM. Recently, a comprehensive meta-analysis review by Yang et al. [36] revealed that VM is associated with not only a worse prognosis in more than ten different tumor types, but also with cancer differentiation, lymph node metastasis, distant metastasis, and the TNM stage. These results suggest that developing strategies against the VM would be a promising therapeutic approach to solid tumors. 

For several years, VM has only been suggested to be associated with tumor metastasis and progression, since dissecting the in vivo hemodynamic function of VM demanded imaging technical assistance. Shirakawa et al. [37] used time-coursed dynamic micro-magnetic resonance angiography (MRA) analysis with an intravascular macromolecular MRI contrast agent [G6D-(1B4M-Gd)256] to investigate the hemodynamics of VM and angiogenesis of the WIBC-9 tumor, demonstrating the existence of a connection between VM and angiogenesis. Interestingly, the published literature and our own study found that sunitinib and even other TKIs that can inhibit angiogenesis might eventually promote VM formation, implicating the need for investigating combination therapies.

The miRNAs are prevalent, important post-transcriptional regulators of gene expression that act by direct base pairing to target the 3′ UTR of messenger RNAs [38,39]. Recently, miRNA activity has been shown to be affected by the presence of miRNA sponge transcripts, the so-called competing endogenous RNA. In addition to LncRNAs, circRNAs represent another promising entity that has been reported as a miRNA sponge. The circRNAs, unlike linear RNAs, form a covalently closed continuous loop and are highly represented in the eukaryotic transcriptome [40]. Recent studies have discovered thousands of endogenous circRNAs in mammalian cells, yet many of them have no identified function(s). Thomas B et al. [41] reported that CDR1-AS, or ciRS-7 (circRNA sponge for miR-7), completely resistant to miRNA-mediated target destabilization, was able to sponge miR-7 and strongly suppress its activity. In RCC, VE-cadherin was partially restrained by tumor suppressor miRNA-125-5p targeting at the 3′ UTR of VE-cadherin mRNA.

Sunitinib has been applied in the treatment of RCC for more than a decade, but the limited therapeutic effect impels research into the development of improvements in treatment efficacy. Emerging evidence indicates that angiogenesis and immunosuppression frequently occur simultaneously in response to tumor microenvironment crosstalk. Accordingly, strategies combining anti-angiogenic therapy and immunotherapy seem to have the potential to tip the balance of the tumor microenvironment and improve treatment response [42]. In recent years, novel TKIs targeting the VEGF signal pathway, such as axitinib, have been advocated to be used in combination with immune checkpoint inhibitors (ICIs) in high-risk metastatic RCC [43]. Yet, the improvement is still limited considering primary and secondary resistance are also commonly seen. Therefore, we tried to explore other alternative treatments that would also complement anti-angiogenesis therapy and improve survival. In our study, we found that sunitinib and axitinib treatment induced the expression of ERβ, which is an important nuclear receptor and an oncogene in RCC. ERβ promotes VM formation by up-regulating the VE-cadherin through an indirect post-transcriptional mechanism. Interestingly, ERβ binds to the promoter of the host gene of circDGKD, increases DGKD transcription, and consequently elevates the level of circDGKD expression. The circDGKD, with its limited size, functions as a specific strong sponge for miRNA-125-5p. 

In our study, miR-125-5p, including miR-125a-5p and miR-125b-5p, which is an RCC tumor suppressor miRNA evidenced by using GEO RCC dataset (GSE37989), targets and is strongly suppressed by the existence of circDGKD. Silencing the circDGKD demonstrated a similar efficacy of disrupting the VM formation capability of the RCC cell via releasing miR-125-5p and consequently decreasing VE-cadherin.

Currently, even the first-line anti-angiogenic treatments, including sunitinib or bevacizumab, fail to suppress VM, and even induce it. Targeting ERβ not only decreases the growth and proliferation of RCC, but also strongly debilitates the formation of VM, which might provide additional benefits for anti-angiogenic therapy.

## 4. Materials and Methods

### 4.1. Cell Culture 

The human RCC cell lines, including 786-O, A498, Caki-1, SW-839, and HEK 293T cells, were purchased form American Type Culture Collection (ATCC, Manassas, VA, USA) and cultured in Dulbecco’s Modified Eagle’s Media (Invitrogen, Grand Island, NY, USA) supplemented with 10% fetal bovine serum (FBS), penicillin (25 units/mL), streptomycin (25 g/mL), and 1% L-glutamine, in 5% CO_2_ humidified incubator at 37 °C. Before any treatment with estrogen, the cells were starved of estrogen in 5% charcoal-dextran-stripped (CDS) FBS DMEM media for 4 days. For the sunitinib or axitinib effect studies, cells were cultured in culture media with 1 µM sunitinib (SU-11248, Selleck) or 1 µM axitinib (AG-013736, Selleck), respectively.

### 4.2. Tube Formation Assay

Thawed Matrigel at 50 μL was evenly loaded to each well of 96-well plates. After Matrigel solidification in 37 °C, 1.2 × 10^4^ 786-O and A498 cells suspended in serum-free culture media were loaded onto the Matrigel. After 6 h of incubation, polygonal structures were captured in 5 fields under 100× microscope. The tubule numbers and tubule lengths were quantitated by Image J as previously reported [44,45].

### 4.3. RNA Extraction, Reverse Transcription, and Quantitative Real-Time PCR Analysis

For RNA extraction, total RNAs were isolated using Trizol reagent (Invitrogen), and miRNAs were isolated by PureLink^®^ miRNA kit according to the manufacturers’ instructions. Quantitative real-time PCR (qRT-PCR) was conducted using a Bio-Rad CFX96 system with SYBR green to determine the RNA expression level of a gene of interest. Expression levels were normalized to the expression of control, GAPDH (for quantifying mRNA) or 5s RNA (for quantifying miRNA). Primers used in this study are summarized in the Appendix A.

### 4.4. Immunohistochemistry (IHC) and Florescence In Situ Hybridization (FISH)

For IHC, slides were deparaffinized, rehydrated, blocked, and then incubated with primary antibodies (ERβ 1:100, Abcam ab288; CD31 1:100, Abcam ab182981; VE-cadherin 1:100, CST 93467). PAS staining was conducted using Sigma PAS kit according to the manufacturer’s instructions and a PAS+/CD31- channel structure was considered to be VM. For any slides, 20 random fields under microscope at 400× were examined; if any one of them had VM then that tissue sample was considered to be VM (+), otherwise VM (−). For FISH, cells were fixed by −20 °C methanol, dehydrated, and hybridized with biotinylated hybridization DNA probe, which was synthesized by Genepharma (Shanghai, China). The probe targeting circDGKD was tagged with cyanine dye 3 (Cy3) and the miR-125b-5p was tagged with fluorescein amidite (FAM).

### 4.5. Western Blot Analysis

Cells were lysed in RIPA buffer and proteins (30 µg) were separated on 8–10% SDS/PAGE gel and then transferred onto PVDF membranes (Millipore, Billerica, MA, USA). After blocking membranes, they were incubated with appropriate dilutions of specific primary antibodies (ERβ, Abcam ab288; VE-cadherin, CST 93467; GAPDH, Abcam ab8245; β-actin, Abcam ab8277). The blots were incubated with HRP-conjugated secondary antibodies, and then visualized using ECL system (Thermo Fisher Scientific, Rochester, NY, USA).

### 4.6. Luciferase Reporter Gene Assay

The 2 kb promoter region of DGKD was constructed into pGL3-basic vector (Promega, Madison, WI, USA). VE-cadherin 1.6 kb 3′ UTR was cloned into the psiCHECK-2 vector construct (Promega) downstream of the Renilla luciferase ORF. Cells were plated in 24-well plates and the plasmids were transfected using Lipofectamine (Invitrogen) according to the manufacturer’s instructions, and pRL-TK was used as internal control for pGL3. After 24 h transfection, cells were lysed with lysis buffer and assayed using the Dual-Luciferase Reporter Assay System (Promega) according to the manufacturer’s instructions. Site-directed mutations were designed according to the Quick Change^®^ Primer Design Protocol (http://sevierlab.vet.cornell.edu/resources/Stratagene-QuikchangeManual.pdf (accessed on 4 February 2020)).

### 4.7. Chromatin Immunoprecipitation Assay (ChIP)

Cell lysates were precleared sequentially with normal rabbit IgG (sc-2027, Santa Cruz Biotechnology) and protein A-agarose. Anti-ER β antibody (5.0 µg) was added to the cell lysates and incubated at 4 °C overnight. For the negative control, IgG was used in the reaction. Specific primer sets designed to amplify a target sequence within the DKGD promoter are listed in Appendix A. PCR products were identified by agarose gel electrophoresis.

### 4.8. RNA Construction and Transduction

The shRNAs were constructed using pLKO.1 vector according to the Addgene TRC Cloning Protocol on the website (http://www.addgene.org/tools/protocols/plko/#B (accessed on 4 February 2020)), between the EcoR1 and Age1 cutting site. The miRNA mimics were constructed between Mlu1 and Cla1 cutting site. Relevant cloning primers were listed in Appendix A. The shRNAs and miRNA mimics were co-transfected with the PAX2 and PMD2G packaging plasmid in HEK 293T cells using the standard calcium phosphate transfection method. After 36 h co-transfection, supernatants were collected and incubated with cells to be infected for 24 h in the presence of polybrene (2.5 μg/mL). After infection, puromycin (1.5 μg/mL) was used to select stably transduced cells. 

### 4.9. Orthotopic Subrenal Capsule Implantation

786-O-Luc-Scr vs 786-O-Luc-shcircDGKD cells (1 × 10^6^) pretreated with vehicle or sunitinib (1 μM) were injected into the left renal capsule of 6-week-old female athymic nude mice (NCI) (*n* = 8 mice per group). Then, mice were treated every other day with/without intraperitoneal injection of sunitinib (20 mg/kg). The tumor sizes and metastatic foci were monitored using the non-invasive in vivo imaging system (IVIS) every two weeks. CD31/PAS double staining was conducted on tumor xenografts to identify VM structures. Studies on animals were conducted with approval from the Animal Research Ethics Committee of the University of Rochester Medical Center.

### 4.10. Statistical Analysis

Data are expressed as mean ± SEM from at least 3 independent experiments with data points in triplicate. Statistical analyses involved Student’s t test, one-way ANOVA, and log-rank (Mantel–Cox) test with SPSS 22 (IBM Corp., Armonk, NY, USA) or GraphPad Prism 6 (GraphPad Software, Inc., La Jolla, CA, USA). *p* < 0.05 was considered statistically significant.

## 5. Conclusions

In this study, we found that ERβ regulates VM formation in RCC, firstly by transcriptionally increasing circDGKD expression, then by acting as a sponge for miR-125-5p, and finally leading to the reduced VE-cadherin, whose 3′ UTR is targeted by miR-125-5p. Targeting ERβ to reduce VM formation in combination with the current anti-angiogenesis treatment might generate synergistic effects in starving the tumor. 

## Figures and Tables

**Figure 1 cancers-14-01639-f001:**
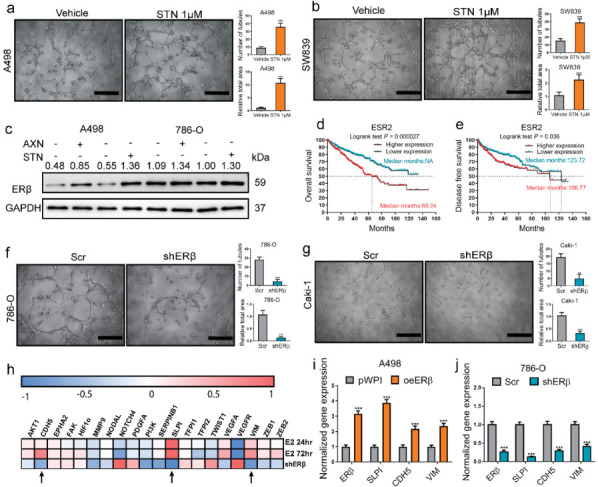
VM formation in RCC induced by sunitinib treatment via ERβ up-regulation could be blocked by silencing VE-cadherin. (**a**,**b**) Matrigel culture system for tube formation assay (images, left) and quantification (right). After sunitinib (STN) treatment (1 μM) for one week, the A498 (**a**) and SW839 (**b**) cells demonstrated increased ability to form polygonal structure both by morphology recognition and by tubule number and area quantification. (**c**) Western blot showed increased levels of ERβ in both 786-O and A498 cells after sunitinib (1 μM) treatment and axitinib (1 μM) treatment. (**d**,**e**) TCGA data analysis between ERβ (ESR2) expression and ccRCC patients’ overall (**d**) and disease-free survival (**e**). (**f**,**g**) Knocking down ERβ in 786-O (**f**) and Caki-1 (**g**) cells severely debilitated their ability to form tubules; quantifications are on the right. (**h**) RNA-Seq analysis of VM-related genes after E2/shERβ treatment. (**i**,**j**) QPCR confirmation of SLPI, VIM, and CDH5 after overexpressing of ERβ (oeERβ) in A498 (**i**) cells or knocking down ERβ (shERβ) in 786-O (**j**). (**k**) Comparison of effect of silencing SLPI (shSLPI), VE-cadherin (shCDH5), and Vimentin (shVIM) in inhibiting VM. (**l**) Representative images of ERβ (upper panels), VE-cadherin (middle panels), and PAS/CD31 (lower panels) IHC double staining in ccRCC samples. Black arrow: VM. (**m**) VE-cadherin positive ratio comparison between ERβ (−/±) or (+/++) tumors of ccRCC patients. (**n**) VM positive ratio comparison between ERβ (−/±) or (+/++) tumors. (**o**) Knocking down VE-cadherin (shCDH5) interrupted the sunitinib-enhanced tube formation. Scale bar, 20 μm, quantifications on the right. Data is presented as mean ± SEM. ** *p* < 0.01, *** *p* < 0.001.

**Figure 2 cancers-14-01639-f002:**
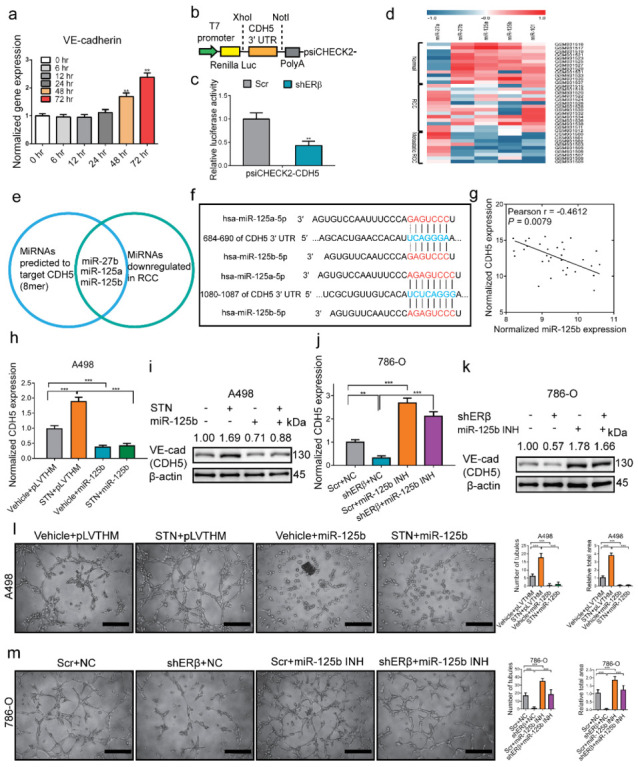
The miRNA-125-5p targets the 3′ UTR of VE-cadherin and is involved in mediating ERβ’s effect on VE-cadherin. (**a**) The qPCR detection of VE-cadherin mRNA at different time points after 10 nM E2 treatment. (**b**) Diagram of constructing the 3′ UTR of VE-cadherin (CDH5) using psiCHECK2 vector. (**c**) Knocking down ERβ (shERβ) decreased psiCHECK2-CDH5 luciferase activity. (**d**) The miRNA heatmap of clinical RCC samples (GSE37989) for screening key miRNAs targeting VE-cadherin (CDH5). Red stands for higher expression and blue stands for lower expression. (**e**) Algorithm for selecting potential tumor suppressor miRNA targeting VE-cadherin. (**f**) Binding site information of miR-125-5p with the 3′ UTR of VE-cadherin (CDH5) from Targetscan. (**g**) Correlation between miR-125b-5p and VE-cadherin expression in clinical ccRCC samples. (**h**–**m**) Interruption assays of miR-125b-5p in regulation of VE-cadherin (CDH5) and VM formation. Overexpression of miR-125b-5p in A498 cells interrupted sunitinib (STN)-induced VE-cadherin (CDH5) mRNA (**h**) and protein (**i**), and interrupted VM formation (**l**). Treating 786-O cells with the miR-125b-5p inhibitor (INH) rescued shERβ-decreased VE-cadherin (CDH5) mRNA (**j**) and VE-cad protein (**k**), and rescued VM formation (**m**). Scale bar, 20 μm. For (**l**,**m**), quantitations are on the right. Data is presented as mean ± SEM. ** *p* < 0.01, *** *p* < 0.001.

**Figure 3 cancers-14-01639-f003:**
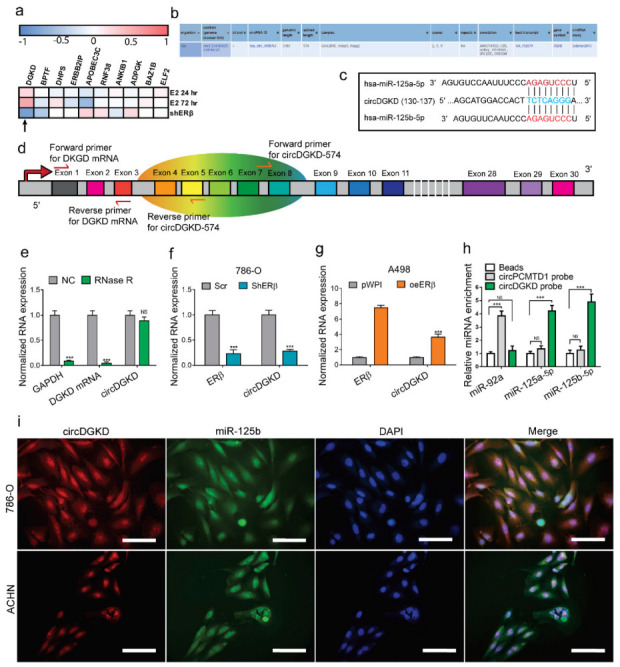
ERβ increases circDGKD that sponges miRNA-125-5p in RCC. (**a**) RNASeq analysis of the gene expression of the top 10 potential circRNA candidates that might sponge miR-125-5p. (**b**) Detailed information about circDGKD retrieved from Circbase. (**c**) Binding site information of miR-125-5p with the circDGKD from Starbase. (**d**) Schematic diagram demonstrating the structure of circDGKD and qPCR primers for circDGKD. (**e**) QPCR confirmation of circDGKD after RNase R digestion. (**f**) Knockdown of ERβ-reduced circDGKD in 786-O cells. (**g**) Overexpression of ERβ-increased circDGKD in A498 cells. (**h**) RNA pull-down assay using biotinylated probe specifically targeting junction region showed that miR-125a-5p and miR-125b-5p were highly specifically enriched in circDGKD, but not in circPCMTD1 (negative control) anti-sense probe pull-down product. The binding of miR-92a to circPCMTD1 was applied as control. (**i**) Fluorescence in situ hybridization (FISH) assay using probe targeting circDGKD tagged with Cy3 (red) and exogenous miR-125b-5p tagged with FAM (green) showed co-localization of circDGKD and miR-125b-5p in 786-O (upper row) and ACHN (lower row) cells. Scale bar, 10 μm. Data are presented as mean ± SEM. *** *p* < 0.001. NS: not statistically significant.

**Figure 4 cancers-14-01639-f004:**
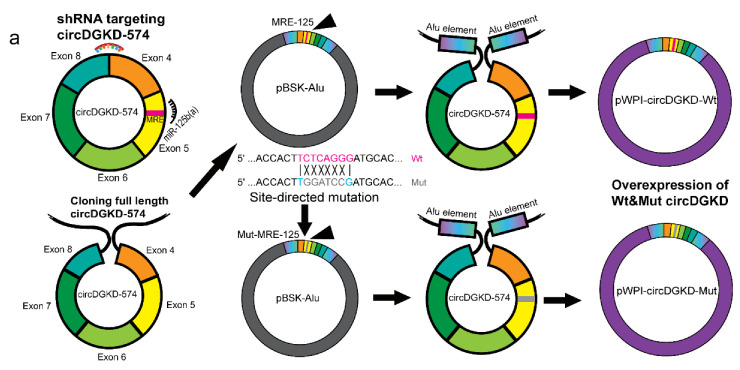
The circDGKD mediates the ERβ-regulated VE-cadherin and VM formation in RCC. (**a**) Schematic diagram of constructing shRNA plasmid of circDGKD targeting the junction region, and wild-type (Wt)/MRE-125 mutant (Mut) circDGKD overexpression plasmid by adding Alu element on the flank. (**b**) Knockdown of circDGKD (shcircDGKD) decreased VE-cadherin level in 786-O cells. (**c**) Overexpression of circDGKD increased VE-cadherin in A498 cells. (**d**) Knockdown of circDGKD and overexpression of miR-125b-5p both decreased psiCHECK2-CDH5 luciferase activity, whereas knockdown of circDGKD, on the basis of overexpressed miR-125b-5p, did not further decrease the luciferase activity. (**e**) Overexpression of Wt circDGKD, but not the circDGKD with MRE-125 binding site Mut, significantly increased the luciferase activity of psiCHECK2-CDH5, in the absence or presence of miR-125. (**f**–**k**) Interruption assays of ERβ and circDGKD regulation of VE-cadherin and VM formation. Overexpression of circDGKD restored shERβ-decreased VE-cadherin in both mRNA (**f**) and protein (**g**), and rescued the shERβ-impaired VM formation in 786-O cells (**j**). Sunitinib (STN) significantly increased VE-cadherin mRNA (**h**) and protein level (**i**) in A498 cells, and boosted the VM formation capacity (**k**), whereas this effect was interrupted by shcircDGKD. Scale bar, 20 μm. For (**j**,**k**), quantitations are at the right. Data are presented as mean ± SEM. ** *p* < 0.01, *** *p* < 0.001. NS: not statistically significant.

**Figure 5 cancers-14-01639-f005:**
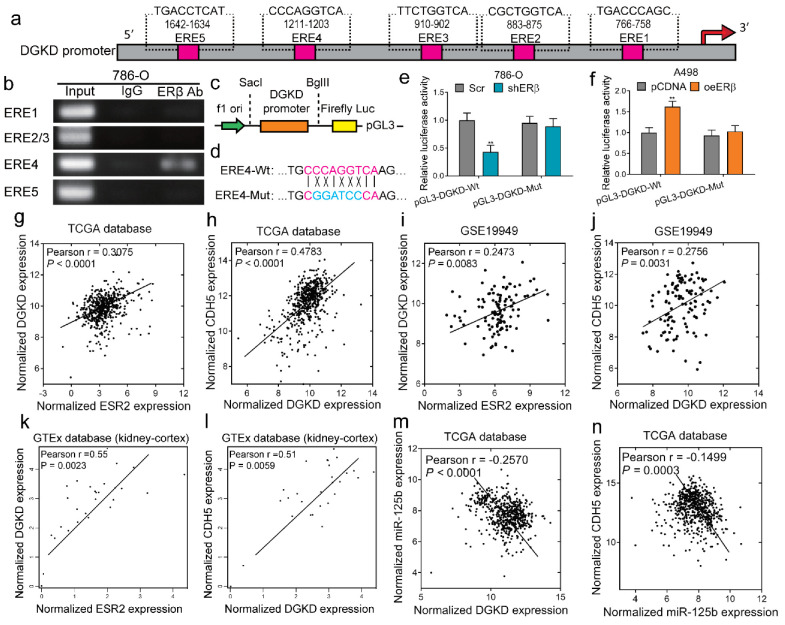
Mechanism dissection showed ERβ binds to the promoter of DGKD and transcriptionally up-regulates DGKD mRNA and circDGKD expression. (**a**) Bioinformatic prediction identified 5 putative ERβ binding sites within 2 kb promoter of DGKD. (**b**) ChIP assay using DNA agarose gel electrophoresis of PCR products revealed ERE4 was the potential binding site. (**c**) Diagram of cloning the 2 kb DGKD promoter into pGL3-basic luciferase reporter vector (pGL3-Wt) between Sac1 and BglII. (**d**) Site-directed mutagenesis of ERE4 by mutating part of the ERE into BamHI cutting site (-GGTACC-). (**e**,**f**) Luciferase assays using wild-type (Wt) or mutant (Mut) pGL3 constructs. In 786-O cells, the shERβ decreased the wild-type luciferase activity significantly, but not the mutant (**e**), and in A498 cells, overexpression of ERβ increased the Wt luciferase activity significantly, but not the Mut (**f**). (**g**–**n**) Public database analysis of relevant correlations. TCGA ccRCC data analysis revealed a positive correlation between the expression level of ERβ (ESR2) and DGKD (**g**), and between DGKD and VE-cadherin (**h**). GEO datasets (GSE19949 from Mayo Clinic) showed a positive correlation between ERβ (ESR2) and DGKD (**i**), and between DGKD and VE-cadherin in ccRCC patients (**j**). GTEx database showed a positive correlation between ERβ (ESR2) and DGKD (**k**), and between DGKD and VE-cadherin (CDH5) (**l**) in non-malignant kidney tissues. TCGA ccRCC data analysis showed a negative correlation between the expression level of DKGD and miR-125b (**m**), and between miR-125b and VE-cadherin (**n**). Data are presented as mean ± SEM. ** *p* < 0.01.

**Figure 6 cancers-14-01639-f006:**
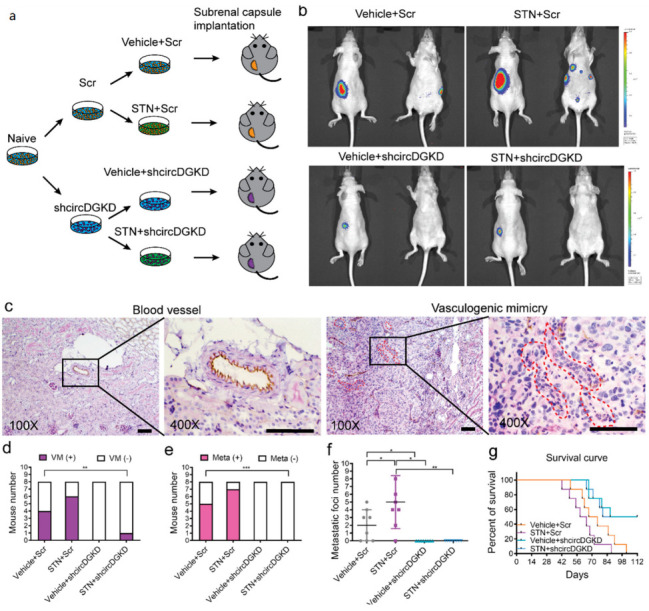
Orthotopic mouse model studies confirmed that shcircDGKD intercepts the sunitinib-pretreatment-induced RCC VM formation, reduces metastases, and improves survival. (**a**) Schematic diagram demonstrating orthotopic mouse model implanted with 786-O-Luc-Scr vs. 786-O-Luc-shcircDGKD cells pretreated with or without 1 μM sunitinib (STN). (**b**) Representative IVIS images of four groups of female nude mice at week 8. (**c**) CD31/PAS double staining comparison between blood vessels in kidney tissue and VM in tumor xenografts. (**d**–**f**) Comparisons of numbers of mice between four groups with VM (+) vs VM (−) (**d**), with positive metastasis [Meta (+)] vs. negative metastasis [Meta (−)] (**e**), and metastatic foci number per mouse (**f**). (**g**) The survival length comparison between the four groups of mice. Scale bar, 100 μm. Data are presented as mean ± SEM. * *p* < 0.05, ** *p* < 0.01, *** *p* < 0.001.

**Figure 7 cancers-14-01639-f007:**
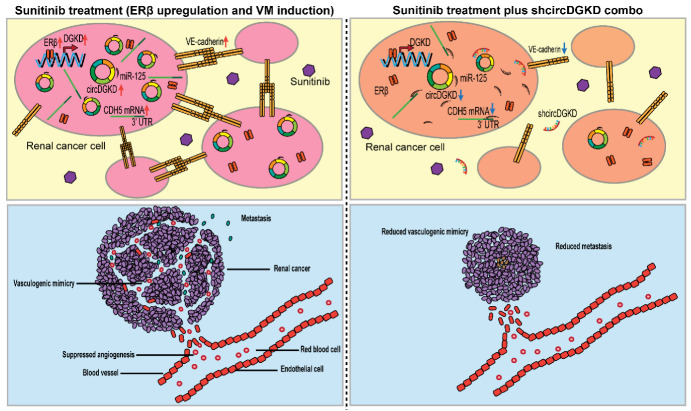
Mechanism diagram illustrating that sunitinib induces VM formation by up-regulating ERβ/circDGKD/miR-125-5p/VE-cadherin axis, whereas sunitinib plus shcircDKGD combo treatment suppresses angiogenesis and VM simultaneously. Red arrows stand for increase and blue arrows stand for decrease.

## Data Availability

The data generated during the current study are available from the corresponding author upon reasonable request.

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
