# Peer review of "Targeting circDGKD Intercepts TKI’s Effects on Up-Regulation of Estrogen Receptor β and Vasculogenic Mimicry in Renal Cell Carcinoma"

_cancers, 2022, doi:10.3390/cancers14071639_

Round 1
Reviewer 1 Report
In this manuscript, the authors demonstrated that estrogen receptor ß participates in sunitinib-induced vascular mimicry. They further demonstrated the role of circDGKD in ER-beta regulation. Overall the research is well designed.
Only some minor concerns are present:
- The resolution of figures must be improved, especially in Fig 1 and 2D. Many words in the figure are hardly to read.
- The vendors of antibodies should be described.
- The description for scale bar in Fig 4i and 4j are missing.
- The scale bar in Fig 6 is missing.
Author Response
Answer to reviewer
- Thank you for your suggestion. In fact the resolution of the original figures is sufficient, yet in this merged Microsoft Word file the images are really blur, far more obscure than our previously uploaded PDF files, which are 300ppi. We suspect it is due to compression by Microsoft Word. In the revised version we uploaded 1200ppi images so that the words are clearer. By in the Microsoft Word version, we could not put so high-resolution figure. Please check the PDF version of figure if there is anything not clear.
- Thank you for your suggestion. The vendors of antibodies of IHC were already provided in the original version of the manuscript. And the vendors of antibodies for WB were added (see page 16 line 461 “they were incubated with appropriate dilutions of specific primary antibodies (ERβ, Abcam ab288; VE-cadherin, CST 93467; GAPDH, Abcam ab8245),.”) in the revised manuscript.
- The description for scale bar in Fig 4i and 4j are now added.
- The scale bar in Fig 6 is now added.

Reviewer 2 Report
Jie Ding et al. uncovered circDGKD to potentially intercepts TKI’s effects on up-regulation of estrogen receptor β. They also focus on vasculogenic mimicry in renal cell carcinoma.
Points to be considered:
- In introduction section a short overview of traditional regarding renal cell carcinoma (RCC) division into three major histopathologic groups-clear cell (ccRCC), papillary (pRCC) and chromophobe RCC (chRCC) would be required from the proposed modern standpoint that can provide useful basis also for the authors'purposes: Marquardt et al. performed a comprehensive re-analysis of publicly available RCC datasets from the TCGA (The Cancer Genome Atlas) database, thereby combining samples from all three subgroups, for an exploratory transcriptome profiling of RCC subgroups (refer to PMID: 33791209)
- While doing matrigel assay a quantification (objective such as with image j would be required or, alternatively, the authors can use already published tools - see PMID: 27940138)
- Did the authors normalize for cell number, viability, proliferation index all the functional assays?
- for all Western blot figures, densitometry readings/intensity ratio of each band should be included; the whole Western blot showing all bands and molecular weight markers should be included in the Supplementary Materials;
-
when discussing the methodology applied for number selection it is not clear for my understanding, how the author selected the sufficient experimental setting. Indeed, in the in vivo experiments, the sample size has been should be calculated in a rigorous way, i.e. by using G*Power software (power of for example 80% and 0.05 statistical level, etc.). Assuming an effect-size of for example, 0.4 with statistical significance of α <;0.05 and a power of 80%. A given 9 mice for each group for a total of 18 mice were estimated. This number should be increased to 20 considering an expected drop-out rate of 10% for the treatment group. Please comment and expand.
- In the discussion section, this reviewer personally misses some information regarding important topics: As is now well known, tumors grow and evolve through constant crosstalk with the surrounding microenvironment, and emerging evidence indicates that angiogenesis and immunosuppression frequently occur simultaneously in response to this crosstalk. Accordingly, strategies combining anti-angiogenic therapy and immunotherapy seem to have the potential to tip the balance of the tumor microenvironment and improve treatment response (please refer to PMID: 32456352).
Author Response
- Thank you for your comment and suggestion. Indeed a brief introduction of different histopathologic types of RCC is needed. So we add a short paragraph, see page 2 line 55: “RCC is divided into three major histopathologic groups-clear cell (ccRCC), papillary (pRCC) and chromophobe RCC (chRCC) , and some rare histopathologic entities, and the histopathology is of crucial relevance for determining treatment strategies including drug sequencing in RCC patients, especially in a metastasized situation.” (refer to PMID: 33791209). And due to the fact that ccRCC constitutes the majority of RCC, and the clinical setting of our pathological studies is primarily focusing on ccRCC, and the TCGA dataset we used is ccRCC patients subtypes, we changed some of the expression in that paragraph from “RCC” to “ccRCC”.
- Thank you for your comment and suggestion. To add automatic intelligent recognition, we tried using the Image J tools to quantitate the tube-formation assay. In the original version we compared only the number of tubules. By using Image J we also compared the total tubule length. In the Methods section we also added tools description, with citation (PMID: 27940138 and 29290800).
After automatic recognition, we keep our original “number of tubules” and add “relative total area”. We think two parameters are enough to show the discrepancy.
- Thank you for this question. As far as we are aware of, the tube formation assay does not require a viability, proliferation index normalization. As described in previously published studies, the cell number quantification is the sole element. In PMID 27940138 Method section, “HUVECs (2×104) and HPAECs (7×104) were seeded on Reduced Matrigel (BD) in EBM-2 with or without Hemilipin2 (50, 100, 250 nM), Hemilipin2 small or large subunit, p-BPB Hemilipin2 (100 nM) followed by the addition of VEGF (50 ng/ml; Sigma Chemical Co.) and FGF-2 (50 ng/ml; Peprotech Inc., Rocky Hill, NJ, USA). After 12 h, the skeletonization of the mesh was followed by the measurement of the relative areas and vessel length in three randomly chosen fields with the EVOS microscope (Thermofisher INC, NYSE, USA) at X200”. Yet you raised a very good question. The tube formation capability might be associated with cell viability, but since there are no published articles reporting this point, and cell viability is not the focus of our study, we didn’t perform such kind of adjustment.
- Thank you for your suggestion. The densitometry readings/intensity ratio of each band are added. For whole Western blot figure, the original membrane of Supplementary Figure S4 b-d blots were cut for the convenience of experiments, because each well of our WB antibody incubation chamber is narrow and short. When we conducted those experiments years ago we didn’t know that we would submit to your journal, and we didn’t know your WB requirement. Otherwise we would definitely keep the membrane as large as possible. We didn’t intentionally keep those markers during our research. And as far as we were aware of, not every journal has this requirement. We guarantee those were definitely not fabricated WB images and we definitely would keep the whole WB membrane issue in mind if we would submit to your journal in the future.
- Thank you for this comment and suggestion. Of course, the sample size of any clinical trial should be calculated in a rigorous way, using either software or formula. When we register clinical randomized trial there is usually such a process demanding calculating sample size, for volunteer recruitment and randomization. However for basic science investigation our institution does not demand such kind of “sample size calculation”. In fact, we also know that more mice, better P value and nicer survival curve. For financial reasons our lab could not support each and every member a large number of mice. And in this study, after observing significant difference in exploratory phase, 8 mice in each group were sufficient to demonstrate the difference.
- Thank you for your comment. We admit that strategies combining anti-angiogenic therapy and immunotherapy could tip the balance of the tumor microenvironment and improve treatment response. This is acknowledged in the field of advanced renal cell carcinoma, especially clear cell renal cell carcinoma (ccRCC). The combination therapy has been written into European Urological Guidelines for years. Confirmed by KEYNOTE clinical trials series, immunotherapy demonstrates its effectiveness in treating ccRCC. In the KEYNOTE-426 study published in NEJM (PMID: 30779529), the investigators found that after a median follow-up of 12.8 months, the estimated percentage of patients who were alive at 12 months was 89.9% in the pembrolizumab–axitinib group and 78.3% in the sunitinib group (hazard ratio for death, 0.53; 95% confidence interval [CI], 0.38 to 0.74; P<0.0001). Median progression-free survival was 15.1 months in the pembrolizumab–axitinib group and 11.1 months in the sunitinib group (hazard ratio for disease progression or death, 0.69; 95% CI, 0.57 to 0.84; P<0.001). The objective response rate was 59.3% (95% CI, 54.5 to 63.9) in the pembrolizumab–axitinib group and 35.7% (95% CI, 31.1 to 40.4) in the sunitinib group (P<0.001).
After all, we didn’t intentionally miss the synergistic effect combining immunotherapy and anti-angiogenic treatment. We did mention in our discussion “In recent years novel TKIs targeting VEGF signal pathway, such as axitinib, are emerging, and are advocated to be used in combination with immune checkpoint inhibitor (ICI) in high risk metastatic RCC. Yet the improvement is still far from satisfactory, since primary and secondary resistance are also commonly seen……”. Thank you for your suggestion, we modified our expression as follows: “Emerging evidence indicates that angiogenesis and immunosuppression frequently occur simultaneously in response to tumor microenvironment crosstalk. Accordingly, strategies combining anti-angiogenic therapy and immunotherapy seem to have the potential to tip the balance of the tumor microenvironment and improve treatment response. In recent years novel TKIs targeting VEGF signal pathway, such as axitinib, are advocated to be used in combination with immune checkpoint inhibitor (ICI) in high risk metastatic RCC. Yet the improvement is still limited, since primary and secondary resistance are also commonly seen. So we try to explore other alternative treatments which would also complement anti-angiogenesis therapy and improve survival.”

Round 2
Reviewer 2 Report
The authors have clarified several of the questions I raised in my previous review. Most of the major problems have been addressed by this revision.